# Lessons learnt about implementing LEGO based therapy (Play Brick Therapy) based on fidelity data and experience from a large school-based randomised controlled trial

Katie Biggs[1]*, Gina Gomez de la Cuesta[2,3], Ellen Kingsley[4], Jean McPherson[2], Fergus Murray[5], Margaret Laurie[2,3], Matt Bursnall[1], Elizabeth Coates[1], Amy Barr[1]

1 Clinical Trials Research Unit, Division of Population Health, University of Sheffield, Sheffield, United Kingdom, 2 Play Included C.I.C, Cambridge, United Kingdom, 3 University of Cambridge, Cambridge, United Kingdom, 4 Child Oriented Mental Health Innovation Collaborative, Leeds and York Partnership NHS Foundation Trust, York, United Kingdom, 5 Autistic Mutual Aid Society Edinburgh, Edinburgh, United Kingdom

* c.e.biggs@sheffield.ac.uk

## Abstract

### Introduction

LEGO® based therapy, a social skills program for autistic children and young people, involves collaborative LEGO® building with adult guidance. This paper examines how well the program was delivered in a recent school randomised controlled trial and explores areas for improvement in implementation. The main trial results are published elsewhere.

### Methods

The I-SOCIALISE trial investigated LEGO® based therapy for autistic children in schools. Researchers recruited 98 schools and randomly assigned them to deliver LEGO® based therapy or usual care. LEGO® based therapy sessions lasted one hour per week for 12 weeks with groups of 3 children. Schools received a 3-hour training session and a manual. Researchers measured fidelity to the LEGO® based therapy programme using self-reported checklists and video analysis. Research team insight and experience of the delivery of the training and intervention are included in this paper.

### Results

LEGO® based therapy was delivered to autistic children in schools with high fidelity according to facilitators and independent reviewers. Most groups (69%) received all 12 sessions, and nearly all groups received the minimum dose of 6 sessions (93%). Sessions typically lasted about an hour and had 1–2 autistic children. Facilitators

**Data availability statement:** Data cannot be shared publicly because of the nature of the informed consent given by participants during the trial. Data are available from the corresponding author in consultation with the co-applicants of the I-SOCIALISE trial for researchers who meet the criteria for access to confidential data. This can be done via the organisational quality assurance team (ctruqa@sheffield.ac.uk).

**Funding:** The author(s) received no specific funding for this work.

**Competing interests:** JM, ML, GGC are employed by Play Included CIC, a not-for profit community interest company. Play Included have funding from the LEGO Foundation. The LEGO® Foundation is a Danish philanthropic organisation who fund charities and not-for-profit organisations to support learning through play. As such, Play Included has links to the LEGO® foundation but not the LEGO® Group (which is a commercial company that sells LEGO bricks). The LEGO group does not have any influence over Play Included. The LEGO Foundation was not involved in any part of the I-SOCIALISE trial or this paper. No commercial LEGO® entities, including the LEGO® group were involved in any part of the I-SOCIALISE trial, except granting special permission for the term LEGO® based therapy Play Brick Therapy to be used in research communications about the I-SOCIALISE trial (now known as Play Brick Therapy).

were mostly teaching assistants with moderate experience in autism. Over 90% of sessions included core elements like group building and social interaction. There were disagreements between facilitators and reviewers on adherence to some program elements like rewards and discussing roles.

## Discussion

LEGO® based therapy was delivered with high fidelity in a large school trial, but there were areas for improvement, such as facilitator training and focus on social interaction for and between children. The authors suggest that facilitators may have been more focused on completing LEGO® builds than on facilitating meaningful social interaction and play between children. Three hours of training may not have been enough to prepare facilitators for their role. The study also did not capture young people's experience of the program, which is important for understanding its effectiveness and impact. Future research should explore how to better measure these aspects and develop a stronger theory of how LEGO® based therapy works.

## Introduction

LEGO® based therapy (now known as Play Brick Therapy. Permission was granted from The LEGO® Foundation to use the term within the context of the I-SOCIALISE study only.) is a child-led social skills programme which focuses on collaborative building of LEGO® models, with playful guidance and facilitation from a trained adult [1,2]. There is a limited, but promising, evidence base showing a positive impact of LEGO® based therapy on the social development of autistic children and young people (henceforth referred to as 'young people') [3,4]. A randomised controlled trial was recently completed in the UK [5,6,7], showing that LEGO® based therapy is a feasible, acceptable [8], and a cost-effective approach [9] to supporting autistic young people in schools to develop their social skills [6,7]. The current paper reflects on the delivery of LEGO® based therapy within this research study, titled I-SOCIALISE (Investigating SOcial Competence and Isolation in children with Autism taking part in LEGO® based therapy clubs in School Environments).

The LEGO® based therapy approach was initially developed in the United States by Dr Daniel LeGoff (2004) [1] after he observed that autistic children attending his psychology clinic naturally engaged and interacted with LEGO® models during sessions. LEGO® based therapy brings young people together around a common goal: to build LEGO® sets or models by working as a team. LEGO® based therapy is a flexible approach where young people can choose to build LEGO® sets or design their own free-style creations in small groups. Through collaborative building, young people naturally use social skills, communicate with each other, and problem-solve. Playful facilitation by the adult is essential to support meaningful social opportunities. Children also develop confidence and creativity, develop a sense of belonging and peer acceptance, develop friendships, and feel pride in their finished models. Previous research has shown that LEGO® based therapy can support social, adaptive and

communication skills in autistic children [10]. Other findings include improved relationships at home between children and their families [11], and higher levels of social play and interaction in other settings, such as the school playground [2].

Fidelity is a multi-faceted concept which broadly refers to the extent to which the delivery of a programme matches the intention according to the relevant governing principles [12,13]. In investigating fidelity, it is possible to explore the relationship between theoretical and practical elements of programme, to better understand the feasibility of programme delivery within everyday settings, and to provide insight into programme outcomes and experiences [14].

In this article, we report on the fidelity of LEGO® based therapy within a randomised controlled trial and reflect on training and implementation of LEGO® based therapy in school settings (see references for the protocol [5] and trial findings [6], and the NIHR report [7] for more details). The trial found a small but non-clinically significant effect on the participants' social skills and this paper explores potential reasons for the small effect based on the fidelity of the training and delivery of the intervention.

The research questions for this article are:

1. How did the implementation of the training and delivery of LEGO® based therapy in school settings impact on the results of the trial?

2. What lessons can we learn to improve the implementation of the programme?

## The research study methods

### Trial design and delivery

The I-SOCIALISE research trial was funded by the National Institute for Health and Care Research's (NIHR) Public Health Research Programme (grant number: PHR15/49/32; trial registry: ISRCTN64852382) and coordinated by the Child Oriented Mental Health Innovation Collaborative (COMIC) and the University of Sheffield Clinical Trials Research Unit (CTRU). The trial was run in mainstream schools in three cities: Leeds, Sheffield, and York, with contiguous rural areas in the North of England. Ethical approval for this study was obtained from the University of York Research Ethics Committee (17/03/2017), and governance approval was granted by the Health Research Authority (18/HRA/01/01), recruitment ran from October 2017 to March 2019. All adult participants (parents/guardians and education professionals) provided written informed consent, and participating young people gave assent where possible and appropriate; this included specific consent to record intervention sessions.

The trial investigated the clinical and cost-effectiveness of LEGO® based therapy for the social and emotional skills of autistic young people, aged 7–15 years, in school environments, when compared with usual support (defined as any support received from school, general practitioners, or any other community professionals). A cluster randomised controlled trial design was used to randomly allocate recruited schools to either the intervention arm – 12 weekly sessions of LEGO® based therapy (in addition to usual support) – or the control arm of usual support only. For each autistic young person, a caregiver, an associated teacher or teaching assistant (TA), and a facilitator staff member (teacher or TA) to run the LEGO® based therapy sessions (intervention schools only) were also recruited.

The trial found a small but non-clinically significant effect on the participants' social skills as measured by the Social Skills Improvement System (SSIS). This was a difference of 3.74 points between the intervention and control groups (95% confidence interval (CI) −0.16 to 7.63 points, p = 0.06) in the intention- to-treat population; and a difference of 4.23 points, (95% CI 0.27 to 8.19 points; p = 0.036) in the per-protocol population. The trial found the intervention was cost-effective through reduced service use costs and a small but significant increase in quality-adjusted life years (QALYs), and that the intervention was acceptable to children and young people, parent/guardians and intervention facilitators.

In total, 98 schools participated in the I-SOCIALISE trial (see Table 1 for further details), 50 schools were allocated to deliver LEGO® based therapy, with 127 young people and 82 trained facilitators in total; this paper only reports

**Table 1. Schools (autistic pupils) recruited by area, arm and level of schooling.**

| | Intervention group | | Control group | | Total |
|---|---|---|---|---|---|
| | Primary | Secondary | Primary | Secondary | |
| | Schools (pupils)* | Schools (pupils)* | Schools (pupils)* | Schools (pupils)* | Schools (pupils)* |
| Leeds | 18 (27) | 6 (10) | 17 (31) | 4 (7) | 45 (75) |
| York | 13 (54) | 3 (20) | 10 (38) | 5 (27) | 31 (139) |
| Sheffield | 8 (11) | 3 (8) | 9 (14) | 2 (3) | 22 (36) |
| Total | 39 (92) | 12 (38) | 36 (83) | 11 (37) | 98* (250) |

Note: 103 schools were randomised. Of these, 5 withdrew before the intervention and are excluded from the table.

on the intervention delivery and does not include information from the 48 control schools. Participants in both arms completed baseline outcome measures and were followed up 20 and 52 weeks after the school was randomised, completing further outcome measures at each time point [5]. Participants were followed up 20 weeks and 52 weeks post baseline to identify the effects immediately after the intervention (whilst allowing for delays in starting and breaks) and longer-term effects of the intervention. LEGO® based therapy sessions were run with three children allocated to each group. Where schools allocated to LEGO® based therapy had recruited fewer than three autistic pupils, 'additional pupils' were recruited for group participation only, no baseline or outcome measures were completed on these additional pupils. These pupils tended to be young people who may benefit from LEGO® based therapy (e.g., on the waitlist for an autism diagnosis, had a diagnosis of another neurodevelopmental condition, or who would otherwise benefit from a play-based peer group programme), or in a few cases, were participants' friends or thought of as 'good role models' by the facilitators.

### LEGO® based therapy training

Training sessions in LEGO® based therapy were delivered in-person (2018–2019) either by members of the local authority in each area, who had expertise in autism, or by members of the study team where local authorities were unable to provide training. All team members who provided training to schools were trained in LEGO® based therapy delivery by experts in the field using a 'train the trainer' model. Training sessions were three hours long and were delivered to 'facilitators', the staff members selected by schools to run the LEGO® based therapy sessions. Facilitators were required to have at least some knowledge and experience of autism, collected via a self-report measure where levels were indicated as limited, moderate, sound, or in-depth.

Training sessions consisted of a presentation covering the background of LEGO® based therapy, how to deliver the programme, and some information about the research study including the specific outcome measures each facilitator would be required to complete. A LEGO® based therapy manual was created for the study by field experts and study team members (this is freely available on the I-SOCIALISE study page at www.comic.org.uk/research/lego). The manual includes instructions on how to run the sessions, editable materials which can be used in the sessions (for example to show who has each role in the LEGO® based therapy sessions), suggestions for free play building activities, and a 'brick naming guide' to support labelling the bricks colours, size and shape if needed. Trainers delivering the sessions were also provided with practical examples of programme delivery. All facilitators were given a copy of both the training presentation and the manual to keep and use to help with their LEGO® based therapy delivery. Follow-up training, additional 'top-up training', supervision or mentoring were not provided in the research trial. Some trainers and facilitators had prior experience of LEGO® based therapy or had previously been trained in a different version of the programme, but no data were collected about this in the research study.

## LEGO® based therapy delivery

Once schools were randomly allocated to the intervention arm of the trial and had received training in LEGO® based therapy, they were able to begin running sessions with their participating young people. Facilitators in participating schools aimed to deliver LEGO® based therapy 12 times over a period of 12 weeks for one hour per week (or at least 45 minutes), with an expectation that school holidays and pupil absence would cause delays with programme completion, or lead to having more than one session a week to catch up. All materials needed to run sessions were provided by the study team (e.g., LEGO® sets of varying difficulty, trays on which to build, and loose LEGO® bricks, Minifigures®, and base plates for freestyle building). It was requested that facilitators identify a quiet and consistent room within school in which to run the sessions, though this was not always possible due to busy school schedules. The LEGO® sets (materials) included a range of difficulty levels from LEGO® Juniors sets (age range 4–7 years) with typically fewer than 100 pieces through to more complex sets (age range up to 12 years) with typically over 150 pieces. This range was intended to cover different ability levels as well as different ages. The training content and intervention delivery were similar/expected to be the same for all ages.

Fidelity to the LEGO® based therapy manual was measured via both self-report and independent analysis of a sub-sample of video recorded sessions. Further details of the fidelity results are reported in Wright et al 2023b but does not include the analysis or discussion in this paper. Wright et al 2023b includes details of the number of sessions delivered and the components delivered based on the facilitator checklist and the independent assessors, and this paper includes additional detail on specific findings from the fidelity assessment, information about the facilitators and discussion around how fidelity may have impacted on the findings.

Facilitator self-reported fidelity was completed after each session using a bespoke checklist designed for the research trial. This was modelled from an existing checklist [15]. The checklist contained 17 items which asked facilitators about encouragement of young people's social interactions (4 items), structural quality of the environment (e.g., setting and resources; 5 items), and specific elements of the programme (e.g., introductions and greetings, group rules, model completion and display; 8 items). Responses included three options: "yes – this item was delivered in the session", "no –was not delivered in the session this time", and "not applicable or not observed".

As part of the trial, the team were required to decide what 'core' aspects of LEGO® based therapy had to be delivered as a minimum for it to meet the definition of LEGO® based therapy and could therefore be included in 'per protocol analysis' [16]. Four items of the bespoke fidelity checklist which focus on the encouragement of young people's social interactions were considered core to the programme for the trial. This is based on the theory behind LEGO® based therapy and its intention as a programme to support social interaction [6,7].

In addition to the self-assessed fidelity, a sub-sample (N=63) of LEGO® based therapy sessions were video recorded to facilitate independent analysis. This was a pragmatic fidelity assessment. To be eligible for this part of the study, all group participants and facilitators needed to consent to be video recorded, so where that was in place, the study team approached the relevant schools to organise video recording; we did not assess the representativeness of the schools that participated in the independent fidelity assessment. Fidelity was analysed by session to understand fidelity over the course of the 12 weeks, and it was deemed more logical to approach the analysis in this way. To provide an independent assessment of fidelity, the recorded sessions were independently reviewed and rated against the fidelity checklist by an independent trained researcher (author AB). A sub-sample of 16 (25%) sessions were double rated by two senior researchers (author GGC and Chief Investigator of the I-SOCIALISE trial BW) and any discrepancies between the two raters were discussed and a consensus reached with a third trained researcher (author EC) to ensure accuracy of rating.

## Analysis

For this article, we are reporting on the data collected in the trial related to training facilitators in the intervention, self-reported fidelity for all intervention sessions (n=749) and a sub-sample (n=63) of sessions independently assessed for fidelity. All analyses are descriptive and were not planned for in the original trial protocol.

## Results

### LEGO® based therapy delivery

In total, 749 sessions of LEGO® based therapy were delivered across all schools allocated to receive LEGO® based therapy in the study. All 12 required sessions of LEGO® based therapy were received by 69% of groups. At least six sessions, defined as the minimum dose as per the study's protocol, were received by 93% of groups. Table 2 shows the number of autistic young people in the groups, where most (51%) primary school sessions only had 1 autistic young person in the group, and most (58%) secondary school sessions had 2 autistic young people in the group.

### Facilitator experience

We collected data from 82 facilitators regarding their role and experience in relation to working with children with autism at baseline (though not all went on to lead or deliver a session) and the data is reported below. Table 3 shows facilitator self-reported knowledge and experience of autism, with most facilitators reporting this as 'moderate'.

Table 4 shows the roles of the facilitators involved in delivering LEGO® based therapy, with most of the staff reporting they were a teaching assistant.

Table 5 shows the number of support programmes for autistic pupils previously delivered by the facilitators, with 10 facilitators having previous experience of some type of "LEGO®-based therapy". Forty-two facilitators had experience of delivering at least one type of support, and 40 facilitators had not delivered any programmes for autistic young people prior to the trial.

### Self-reported fidelity

Overall completion of the fidelity checklists was high (99%), with only seven missing checklists from the total of 749 sessions completed during the trial. The proportion of items self-rated as delivered by facilitators ("Yes" responses) across all sessions was 91%. In total, 98.6% of sessions included the core four items of the fidelity checklist, showing good self-reported adherence to the core definition of LEGO® based therapy.

For 71 facilitators (those who delivered the intervention in the trial and completed self-report forms), we have looked at the level of autism knowledge and their experience delivering interventions for autism against their self-report ratings.

**Table 2. Number of autistic young people in session.**

| Number of autistic young people in the group | Primary school sessions, n (%) | Secondary school sessions, n (%) |
|---|---|---|
| 1 | 292 (51) | 30 (18) |
| 2 | 157 (27) | 96 (58) |
| 3 | 125 (22) | 40 (24) |

This table has been adapted from Wright et al 2023 (PHR report) with permission.

**Table 3. Facilitator (N = 82) reported knowledge and experience of autism.**

| Facilitator reported knowledge and experience of autism | Number of facilitators (%) |
|---|---|
| In-depth | 14 (17.1) |
| Sound | 24 (29.3) |
| Moderate | 36 (43.9) |
| Limited | 8 (9.8) |

**Table 4. School role for the facilitators (N = 82) delivering the trial intervention.**

| School role | Number of facilitators (%) |
|---|---|
| SENCO | 6 (7.3) |
| Assistant to SENCO | 1 (1.2) |
| Teacher/ educator | 2 (2.4) |
| Teaching assistant | 50 (61.0) |
| Special needs assistant | 1 (1.2) |
| Emotional Literacy Support Assistant (ELSA) | 1 (1.2) |
| Higher Level Teaching Assistant (HLTA) | 8 (9.8) |
| Lead autism practitioner | 1 (1.2) |
| Learning mentor | 8 (9.8) |
| LSA | 1 (1.2) |
| Nursery officer/nurture | 1 (1.2) |
| Pastoral care officer | 1 (1.2) |
| Senior member of staff | 1 (1.2) |

**Table 5. Facilitator (N = 82) experience delivering autism specific school-based interventions.**

| Number of interventions delivered previously | Number of facilitators delivering (%) |
|---|---|
| 0 | 40 (48.8) |
| 1-3 | 22 (26.8) |
| 4-6 | 17 (20.7) |
| 7-9 | 1 (1.2) |
| 10-12 | 2 (2.4) |

Table 6 below shows the average fidelity ratings across the 4 core items, and for all items by their level of experience. This is an average rating and does not account for trends over time. The self-report rating on all items does not appear linked to the facilitators experience, with moderate and sound experience leading to the highest self-report ratings and those delivering 0 interventions scoring higher on the self-report ratings as well. However, for the 4 core items on the checklist, there appears to be an increase in self-reported fidelity with increased knowledge of autism, but there was no clear pattern with increased numbers of interventions delivered previously.

**Table 6. Facilitator (N = 71) self-report fidelity ratings by knowledge of autism and experience delivering autism interventions.**

| Measure of previous experience** | | N (%) | Self-reported fidelity rating on all items* | | | Self-reported fidelity rating on 4 core items* | | |
|---|---|---|---|---|---|---|---|---|
| | | (N=71) | Mean (SD) | Median (IQR) | Min., Max. | Mean (SD) | Median (IQR) | Min., Max. |
| Facilitator reported knowledge and experience of autism | Limited | 7 (10%) | 0.92 (0.07) | 0.91 (0.86, 0.99) | 0.82, 1.00 | 0.97 (0.05) | 1.00 (0.94, 1.00) | 0.88, 1.00 |
| | Moderate | 32 (45%) | 0.95 (0.05) | 0.97 (0.92, 0.99) | 0.82, 1.00 | 0.98 (0.03) | 1.00 (0.98, 1.00) | 0.85, 1.00 |
| | Sound | 20 (28%) | 0.95 (0.04) | 0.95 (0.93, 0.98) | 0.88, 1.00 | 0.98 (0.06) | 1.00 (0.98, 1.00) | 0.75, 1.00 |
| | In-depth | 12 (17%) | 0.94 (0.04) | 0.94 (0.90, 0.97) | 0.87, 1.00 | 0.99 (0.02) | 1.00 (0.99, 1.00) | 0.94, 1.00 |
| Number of interventions delivered previously | 0 | 35 (49%) | 0.95 (0.05) | 0.97 (0.92, 0.99) | 0.82, 1.00 | 0.98 (0.03) | 1.00 (0.98, 1.00) | 0.85, 1.00 |
| | 1-3 | 18 (25%) | 0.94 (0.04) | 0.94 (0.92, 0.97) | 0.86, 1.00 | 0.99 (0.03) | 1.00 (0.98, 1.00) | 0.88, 1.00 |
| | 4+ | 18 (25%) | 0.94 (0.05) | 0.94 (0.90, 0.99) | 0.87, 1.00 | 0.98 (0.06) | 1.00 (0.98, 1.00) | 0.75, 1.00 |

*Each facilitator rated 17 aspects of delivery for between 1 and 12 sessions using responses: adhered to (Yes); not adhered to (No) or not relevant (n/a). A score for each facilitator was calculated as the total number of Yes answers divided by the total number of Yes or No answers across all sessions rated. ** Collected at baseline following facilitator consent

## Independent fidelity assessment

In total, 63 sessions from 22 schools (and 22 facilitators) were video recorded and independently rated, representing 9% of all LEGO® based therapy sessions implemented in the study. Ten of the 22 groups were from Leeds schools: 8/22 from Sheffield and 4/22 from York. Most LEGO® based therapy sessions recorded were from primary schools (18/22) which reflects the abundance of primary schools recruited to the study. The average duration of session recordings included in this sub-study was 55 minutes (range: 30–78 minutes). The recorded LEGO® based therapy sessions included 20 early sessions (sessions 1–4 of 12), 22 middle sessions (sessions 5–8) and 21 later sessions (sessions 9–12). No first sessions were recorded with the number of all subsequent sessions varying from third to ninth session.

Overall, fidelity within this sub-study was very high: 89.5% for facilitator self-report ratings and 83.1% for independent ratings. The level of inter-rater reliability was high at 84%, and all discrepancies between the two raters were discussed. Consensus was then achieved through discussion with the third researcher.

Again, we looked at the independent fidelity ratings according to the facilitators experience (N = 15, lower than the number delivering due to missing data or because there was more than one facilitator present). Table 7 shows the independent fidelity ratings for all recorded sessions we could match with a unique facilitator by the level of experience of the facilitator, and this shows that on both experience measures, the ratings were slightly higher where facilitators were more experienced.

## Descriptive analysis of fidelity

The 17 items of fidelity were split into three separate groups. Trends were investigated in each group to explore which aspects of LEGO® based therapy were most consistently implemented during the research trial. These groups were *Facilitation of Social Interaction,* which included the 4 items 'core' to LEGO® based therapy, *Structural Quality,* which included 5 items, and *Adherence to Programme,* which included 8 items. Fidelity rating by item and points of disagreements between facilitator and independent ratings are reported in Table 8. The table shows whether there was agreement on the checklist items between the facilitator reported fidelity and the independent assessment, and whether they had agreed the item was present (yes), or not (no). Most items showed high levels of agreement. For example, 100% of respondents agreed that there was a sufficient adult: child ratio (at least 1:3). Similarly, 96.8% agreed that the children displayed an understanding of LEGO®-based therapy. In terms of adherence to the intervention, only 41.6% of respondents agreed that children use appropriate greetings and 69.8% of respondents agreed that they described what would happen in the session. Regarding the dismantling of models, only 66.6% of respondents indicated that warning and explanation were given, and children were allowed to dismantle the models themselves, though it is important to note that not all sessions would have had a model to dismantle. Where there was a disagreement, this tended to be the result of facilitators reporting delivery of a component that was not observed independently in the recorded footage (9%).

**Table 7.  Facilitator (N = 15) independently reported fidelity ratings by knowledge of autism and experience delivering autism interventions.**

| Measure of previous experience** | | n (%) | Independently reported fidelity* | | |
|---|---|---|---|---|---|
| | | (N = 15) | Mean (SD) | Median (IQR) | Min., Max. |
| Facilitator reported knowledge and experience of autism | Limited/Moderate | 7 (47%) | 0.78 (0.16) | 0.79 (0.63, 0.88) | 0.53, 1.00 |
| | Sound/In-depth | 8 (53%) | 0.85 (0.05) | 0.85 (0.82, 0.89) | 0.77, 0.91 |
| Number of interventions delivered previously | 0 | 10 (67%) | 0.79 (0.13) | 0.82 (0.74, 0.87) | 0.53, 1.00 |
| | 1-3 | 5 (33%) | 0.86 (0.06) | 0.88 (0.86, 0.90) | 0.77, 0.91 |

*Same calculations as Table 6. ** Collected at baseline following facilitator consent

**Table 8. Summary of fidelity agreement ratings for each item.**

| Theme | Item | Agreed - response = 'yes' (%) | Agreed - response = no (%) | Disagreement (%) | Missing |
|---|---|---|---|---|---|
| *Facilitation of social interaction* | Were the children encouraged and supported to interact and cooperate with each other? | 56 (88.8%) | 0 | 7 (11.1%) | 0 |
| | Did each child work with another child or group of children to build a LEGO® model (e.g., using the roles Engineer, Supplier and Builder, or designing their own freestyle models together)? | 61 (96.8%) | 0 | 2 (3.17%) | 0 |
| | Did you highlight positive social interactions with the children? | 50 (79.3%) | 0 | 13 (20.6%) | 0 |
| | Did you highlight some aspect of social interaction (e.g., a point of learning, a challenge, issue or difficulty) and prompt children to problem solve or discuss it with each other? | 43 (69.3%) | 2 (3.2%) | 18 (28.6%) | 0 |
| *Structural quality* | Was there a sufficient adult:child ratio (at least 1:3)? | 63 (100%) | 0 | 0 | 0 |
| | Was the session held in a good space (e.g., quiet, undisturbed, an adequate play surface provided)? | 55 (87.3%) | 1 (1.5%) | 7 (11.1%) | 0 |
| | Was there evidence of group rules that children were aware of? | 47 (74.6%) | 0 | 16 (25.3%) | 0 |
| | Did the children display understanding of LEGO® based therapy by engaging in appropriate activities (e.g., building with other children, following their roles, following the rules)? | 61 (96.8%) | 0 | 2 (3.1%) | 0 |
| | Was the LEGO® used age-appropriate and organised coherently? (E.g. in sets according to colour and function, in suitable storage boxes) | 60 (96.7%) | 1 (1.6%) | 2 (3.2%) | 0 |
| *Adherence to intervention* | Did the children use appropriate greetings (saying 'hello' and 'goodbye' and using each other's names)? Or, if not, were they prompted to do so? | 25 (41.6%) | 4 (6.6%) | 34 (51.6%) | 0 |
| | Did you describe what will happen in the session? (e.g., verbally or with the use of a visual timetable) | 44 (69.8%) | 1 (1.5%) | 18 (28.5%) | 0 |
| | Did the children participate in a discussion about who will build what and who will play each role Engineer, Supplier, Builder) | 43 (68.2%) | 5 (7.9%) | 15 (23.8%) | 0 |
| | Were the children in the same physical area as their partners, such as all around a table or all working in the same space on the floor? | 62 (89.4%) | 0 | 1 (1.5%) | 0 |
| | Was there an opportunity for each child to show the other group members what they made, or talk about or play with what they made with others? | 34 (79%) | 1 (2.32%)<br>4 (6.6%) N/A* | 22 (34.9%) | 2 |
| | When a model is complete, is it appropriately displayed or are photographs of the model taken? | 37 (74%) | 5 (10%)<br>5 (7.9%) N/A* | 15 (23.1%) | 1 |
| | If it was necessary to dismantle the models, was there warning and explanation given and were the children allowed to dismantle the model themselves | 16 (66.6%) | 0<br>24 (38.1%) N/A* | 23 (36.5%) | 2 |
| | Was there evidence of a reward (e.g., LEGO® prints, stickers, praise) to promote positive social behaviour, or LEGO® certificates for attaining the different group levels (helper, builder, creator, etc.)? | 33 (54%) | 12 (19.6%) | 16 (26.22%) | 2 |

*N/A was indicated in the items 14, 15a and 15b when the session was thought not to include a complete model.

The denominator for working out the percentages is 63 (the number of sessions that were independently assessed).

## Facilitation of social interaction

Overall, both facilitators and independent raters reported high fidelity to facilitation of social interactions, in particular to having young people work together as a group on LEGO® models and engage in interaction (average fidelity rating across facilitator ratings: 98.8% and independent ratings: 91.2%). There was less agreement on fidelity of facilitators' highlighting positive social interactions and highlighting opportunities for young people to engage in problem solving and managing social interactions – usually with the independent rater disagreeing with the self-reported implementation of an item.

## Structural quality

Overall, fidelity to structural quality regarding LEGO® based therapy sessions was very high (average fidelity rating: 98.7% and average independent rating: 95%). This is likely due to the context of being embedded within a research trial where many factors such as group size and setting are controlled by design. There was more disagreement between facilitators and independent raters on there being evidence of group rules that young people were aware of. However, there were some instances where videos had the start of the session missing, meaning that these were possibly under-reported rather than not implemented.

## Adherence to the programme

It is here that average fidelity and agreement between facilitators and independent raters begins to decrease compared to the other two groups of items (average fidelity rating: 79.1% and average independent rating: 71.7%). The two areas with lower self-reported adherence were: providing a reward at the end of the session, and displaying and/or dismantling completed LEGO® sets when there was an appropriate opportunity. There are also some items where there are higher levels of disagreement between facilitator and independent raters, such as the facilitator describing what will happen in a session, and young people participating in a discussion around the different roles in LEGO® based therapy sessions. Fidelity assessors were informed on one occasion that some rewards were given in a school assembly, and it may be that rewards were given outside the session, and not recorded, again indicating under-reporting of reward giving.

# Discussion

## Findings in context

LEGO® based therapy is a widely used approach to support young people to collaboratively build LEGO® models in a supportive environment, facilitated by a trained adult. There is a growing evidence base for using LEGO® based therapy to support social and emotional skills in autistic young people, with the I-SOCIALISE trial showing small improvements in social skills [6,7], but it is important to consider the fidelity to the programme and whether this could have impacted on the results. This paper reviews the implementation of the training and delivery of LEGO® based therapy in school settings, how this may have impacted on the results, and what lessons can be learnt to improve implementation. We measured fidelity in two ways: through self-report and through independent assessment of 9% of sessions.

Overall, fidelity of LEGO® based therapy was high across the 50 schools randomised to the intervention in the I-SOCIALISE trial. The most consistently implemented items related to the young people building LEGO® models together in a group in a structured way, and items that related more to adults' facilitation of social interactions were found to have lower fidelity in independent evaluations. We identified some key implementation barriers that emerged from the study: facilitator training and understanding of the intervention; focus on task completion over social interaction; challenges in recognising and reinforcing social interactions; contextual factors; and limitations in measurement.

## Facilitator training and understanding of the intervention

Variable Facilitator Experience with Autism and Intervention Delivery: As seen in Table 3, facilitator knowledge and experience of autism was collected with the majority reporting moderate but some reporting limited knowledge and experience. According to a recent report by the National Autistic Society (2023) [17], only 39% and 14% of primary and secondary teachers respectively have received more than half a day's training about autism. Facilitator experience of delivering autism specific programmes was also collected (see Table 5) with just under half of all facilitators reporting no experience of this. This was a self-report tool, however, and some facilitators may not have considered that some of their experiences would be classed as delivering a specific programme (e.g., 1:1 support). No facilitators were excluded from LEGO® based therapy delivery due to their level of knowledge and experience of autism or other programme delivery. This variability

could have impacted their ability to effectively facilitate the sessions and understand the nuances of autistic social interaction. There was some evidence of this when looking at the independent fidelity ratings by facilitator experience, as fidelity ratings were slightly higher for facilitators with more experience (Table 7). Although there was an indication of this in the self-report for the 4 core fidelity items, this was not shown in the self-report ratings across all fidelity items. It should be noted that these are small numbers, and the nature of self-reported fidelity, which could also vary by experience, means we should not draw firm conclusions based on these numbers.

Underestimation of Facilitator Role: The variability in experience may have implications for delivery success and fidelity, especially in the context of some education professionals potentially not fully understanding their role. While facilitators understood the roles for young people within LEGO® based therapy sessions, they were perhaps less clear on their own role as adults facilitating the sessions or less willing to implement guided play.

Reasons for this could relate to the children and adults having previous relationships within the school, and school being a setting in which adult-led activity is often the norm (e.g., Skene *et al.,* 2022, [18]). In LEGO® based therapy the emphasis should be very strongly on child-led sessions with the young people having agency over what they build, who they build with and how they play together.

Insufficient Training Duration: The three-hour training session provided to facilitators was a pragmatic decision based on time barriers encountered by the study team when liaising with schools and local authorities. This limited time, which meant some content was condensed or not covered, might have prevented a deeper understanding of the facilitator's role. Training also had to include some information around the trial, reducing the training content further. Time barriers further meant that additional training or supervision of delivery of LEGO® based therapy was not available to facilitators. A lack of understanding around active, experiential role-playing could have impacted on playful facilitation in LEGO® based therapy sessions. Some facilitators might also have been concerned that too much involvement on their part might risk undermining young people's intrinsic motivation [19].

It may also be viewed, by professionals and authorities, that a play-based programme such as LEGO® based therapy requires less rigorous training or support to deliver, because it is viewed as "just playing" with children or requires less "hands-on" facilitation or teaching. Often, additional programmes, especially those that focus on social and emotional learning such as LEGO® based therapy, are delivered by teaching assistants and paraeducators. These paraeducators are an invaluable classroom resource – but training and resources provided to deliver such interventions within these roles can vary widely, which can impact on the efficacy and impact of such programmes in school settings [20].

Some facilitators had previous experience of delivering LEGO® based therapy with different aims, for example for speech and language therapy, and others may have had preconceptions about it based on previous experience with similar or adjacent programmes which the time-limited training was unable to combat. Such programmes commonly have a lesser focus on the adult facilitation necessary to scaffold collaborative play and may have had other elements inconsistent with training for this trial (e.g., adults allocating roles, and common misconceptions about LEGO® based therapy such as only one child being allowed to see the building instructions and a dominant focus on descriptive language of the bricks themselves rather than play and socialising). Guided play, where young people are given freedom to explore a learning goal in their own way under meaningful steer from an adult, is consistently found to be beneficial to young people's outcomes in school settings [18]. However, many teachers may not have the resources, capacity, prior experience or training to deliver quality guided play in the classroom [18]. Therefore, greater awareness and attention should be paid by the adult to their role in the LEGO® based therapy sessions – to foster, find and facilitate meaningful social opportunities between young people, and to adopt a young person-centred focus to support the group to problem-solve and collaborate.

## Focus on task completion over social interaction

Another potential explanation for the intervention not being as effective, is that facilitators may have been more focused on the set building and completion, rather than the process of young people spending quality time together and

developing positive relationships with each other. This may have occurred due to the context of a research study where facilitators knew that they would be assessed on number of sessions delivered and criteria such as model or set completion and were therefore driven to complete the sessions and models accordingly. This raises a need for continuous training and ongoing formative reflection of the LEGO® based therapy program in practice, to reinforce to facilitators the importance of highlighting the positive social experiences and play for young people within LEGO® based therapy, beyond the LEGO® model building.

### Challenges in recognising and reinforcing social interactions

In this research study, LEGO® based therapy was implemented with groups of three children per session, with 1–3 autistic young people in the group and other young people invited by the school as required ('additional children'). It should be noted that LEGO® based therapy is a flexible approach which can be implemented in larger groups of young people and to those with broader social and communication needs, such as other neurodivergent young people with ADHD or social anxiety (LeGoff and Sherman, 2006). One fidelity checklist item which had a lower score was facilitators reporting that they pointed out positive social interactions in the group. A possible explanation for this is the double empathy problem, where non-autistic people may have difficulty interpreting the social behaviours of autistic people [21]: neurotypical facilitators may have been less likely to recognise positive moments of interaction for the neurodivergent young people, reflecting findings from other work on neurotypical adults' perceptions of autistic interactions [22].

### Limitations in measurement

This study has found that overall fidelity in LEGO® based therapy was high, which goes in part to explain the small positive findings observed in child outcomes [6,7]. However, there are other elements of facilitation and programme experience not measured in the current study which are fundamental and possibly drive outcomes of LEGO® based therapy and similar programmes. For example, the experience of connectedness with peers can have a positive effect on children's social, emotional, and cognitive development [23]. Having fun, feeling happy and experiencing joy are also fundamental aspects of playful experiences that link to children's learning and development [24]. From the young person's perspective, feeling like they have a say in LEGO® based therapy sessions can increase self-confidence and efficacy, and allow for the development of leadership and other social emotional skills. From the facilitator's perspective, being able exercise a range of facilitation techniques, being responsive to the young people in the session, and taking active roles when needed, can also create a positive and effective environment for young people's development. These programmatic aspects were not directly measured in the fidelity checklist of this study but may be fundamental to the success of LEGO® based therapy and similar programmes to support young people's social and emotional development. As suggested in Wright et al (2023) and Evans and Bond (2021) [6,25], an articulated theory of change would benefit from further testing and refinement from a theory of change to support programme implementation and evaluation. The first element relates to young people's experiences of joy, positive affect, social connectedness or belonging – although hard to measure objectively – which could be argued as being the fundamental underpinnings of bringing young people into a learning through play experience [24]. The extent to which young people had agency and took the lead in sessions – i.e., making decisions around session activities, group rules, deciding when to switch roles, problem-solving – was also not captured directly. Measurement of aspects of learning through play – including the characteristics of playful learning and the implementation of different types of guided play – would have uncovered more about young people's experiences of the programme and helped elucidate the link to positive social outcomes reported elsewhere [6,7].

In summary, the I-SOCIALISE trial observed a small, non-clinically significant improvement in participants' social skills, which we believe is due to challenges in the fidelity of the intervention's delivery, particularly regarding social interaction facilitation. While structural elements like collaborative building were largely adhered to, facilitators, who had variable prior experience with autism and seemed to underestimate their crucial role in guided play, tended to focus more on completing

LEGO® builds rather than actively fostering and reinforcing social interaction and positive relationships among the young people. This was exacerbated by the limited three-hour training duration, which likely prevented a deeper understanding of the facilitator role in supporting social development through play, and challenges in recognising and highlighting positive social interactions for autistic participants. Consequently, the fidelity gaps in facilitating the core social aims of the therapy are considered key contributors to the intervention's limited impact on social skills, underscoring the need for future training to prioritise active guided play and understanding neurodivergent communication styles.

### Limitations of the current paper

This paper reflects the delivery of LEGO® based therapy with the I-SOCIALISE trial, and it was not a pre-planned output of the trial and only 9% of the sessions were independently assessed for fidelity, so most of our findings are based on self-reported fidelity. The main measure of fidelity was by self-report checklists completed by the facilitators, where there is potential for overreporting fidelity [26] and can be unreliable due positivity bias [27]. This was evident when compared to the independent assessments, but agreement between our self-report and independent assessments were generally high and we did not have the resources to independently assess every session. The independent assessment was conducted on a per-session basis, not on a per-facilitator basis. Although 22 facilitators were assessed over two or more sessions throughout the 12-week intervention, this approach makes it challenging to attribute the fidelity findings to specific facilitators' experience or characteristics. This, in turn, makes it difficult to pinpoint individual training needs or areas for improvement for practitioners in future interventions.

LEGO® is a globally recognised toy and there is evidence of cross-cultural differences in Autism [28] but we are unable to comment on this as we did not collect data on county of origin, and the sample was not diverse enough for us to explore cultural differences. However, there has been some evaluation of the intervention in Australia, Denmark, Ireland, Kenya, Malta, Mexico, and the US where it was feasible to deliver and acceptable to children. The qualitative study found positive changes in children's enjoyment of being in a group, confidence, friendship and language development, but there was no impact on anxiety, emotional regulation or understanding of the child's own feelings [29].

Process evaluation, a common practice in complex intervention trials, was included but at a limited scope due to funding restrictions. Although we did look at individual facilitator experiences and how this linked to the fidelity assessment, but this was not resourced for or pre-planned. It's important to note that trials of complex interventions increasingly emphasise using a clear theory of change and logic model to assess intervention effectiveness, helping to identify factors influencing when and why the intervention works, a pre-liminary logic model (S1 Fig) is based on the delivery in the trial and the trial findings.

### Recommendations for trials

There were limitations placed on LEGO® based therapy in the trial that would not be necessary in real-world delivery, which was important to maintain the scientific integrity of the trial. However, researchers should consider how the delivery of LEGO® based (or similar) therapy is impacted by a trial, and what can be done to overcome this. For example, providing facilitators with videos or access to trainers for ongoing support, or ensuring a minimum standard for facilitator experience if training needs to be shortened for a trial. The young person's voice should have been heard more in this trial, in terms of public involvement and in interviews relating to acceptability, and we recommend future research does this.

### Conclusion

This paper looks at the delivery of LEGO® based therapy and how this may have affected the trial findings. We propose areas which are ripe for further training of professionals delivering LEGO® based therapy or similar programmes for autistic young people that may improve the effectiveness of these interventions. There is a need to support facilitators to be active in guided play and to support young people's positive social experiences and interactions. While there is an

important focus on young people building models together as part of the LEGO® based therapy sessions, the sessions also provide positive social experiences for young people, and so facilitation and training must focus more on these aspects, particularly when facilitators do not have experience delivering similar interventions. Further training on neuro-diversity, including neurodivergent communication styles, may be warranted to encourage positive engagement in the LEGO® based therapy sessions, particularly when run by neurotypical facilitators or when they include mixed neurotypical and neurodivergent group members.

## Supporting information

**S1 Fig. Pre-liminary logic model for Play Brick Therapy based on the trial conduct and delivery.**
(DOCX)

## Acknowledgments

We acknowledge and thank the Chief Investigator of I-SOCIALISE, Professor Barry Wright and the full I-SOCIALISE team that successfully undertook and completed the trial: Shehzad Ali, Matthew Bursnall, Tim Chater, Cindy Cooper, Simon Gilbody, Ann Le-Couteur, David Marshall, Kirsty McKendrick, Roshanak Nekooi, Anna Packham, Steve Parrott, Kiera Solaiman, Dawn Teare, Danielle Varley and Han-I Wang.

We would also like to thank all of the children and young people, parents and guardian and teachers that took part in the trial.

## Author contributions

**Conceptualization:** Katie Biggs, Gina Gomez de la Cuesta, Elizabeth Coates.

**Data curation:** Katie Biggs, Ellen Kingsley, Matt Bursnall, Elizabeth Coates, Amy Barr.

**Formal analysis:** Katie Biggs, Gina Gomez de la Cuesta, Ellen Kingsley, Jean McPherson, Fergus Murray, Margaret Laurie, Matt Bursnall, Elizabeth Coates, Amy Barr.

**Funding acquisition:** Katie Biggs, Gina Gomez de la Cuesta, Elizabeth Coates.

**Investigation:** Katie Biggs, Gina Gomez de la Cuesta, Ellen Kingsley, Matt Bursnall, Amy Barr.

**Methodology:** Katie Biggs, Gina Gomez de la Cuesta, Ellen Kingsley, Elizabeth Coates.

**Project administration:** Katie Biggs, Ellen Kingsley, Matt Bursnall, Amy Barr.

**Resources:** Katie Biggs.

**Supervision:** Gina Gomez de la Cuesta, Elizabeth Coates.

**Writing – original draft:** Katie Biggs, Ellen Kingsley.

**Writing – review & editing:** Katie Biggs, Gina Gomez de la Cuesta, Ellen Kingsley, Jean McPherson, Fergus Murray, Margaret Laurie, Matt Bursnall, Elizabeth Coates, Amy Barr.

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
