## [Decision Letter · Decision Letter 0]

26 Nov 2024

Dear Dr. Biggs,

Thank you for submitting your manuscript to PLOS ONE. After careful consideration, we feel that it has merit but does not fully meet PLOS ONE’s publication criteria as it currently stands. Therefore, we invite you to submit a revised version of the manuscript that addresses the points raised during the review process.

We look forward to receiving your revised manuscript.

Kind regards,

Christopher James Hand, Ph.D., M.Sc., M.A., PgCAP

Academic Editor

PLOS ONE

Journal Requirements:

“I have read the journal's policy and the authors of this manuscript have the following competing interests: LEGO® is a registered trademark with a fair use policy which was adhered to throughout the duration of the trial. The LEGO® Foundation have agreed to the use of this term solely for this project and its outputs. Author GGC is a Director of Play Included, a community interest company that offers training and resources for interventions involving play bricks for children. Author MG is employed by Play Included, and JM worked for the company at the time of writing the paper.

Other authors report no conflict of interest.”

We note that one or more of the authors are employed by a commercial company: LEGO

4. In the online submission form you indicate that your data is not available for proprietary reasons and have provided a contact point for accessing this data. Please note that your current contact point is a co-author on this manuscript. According to our Data Policy, the contact point must not be an author on the manuscript and must be an institutional contact, ideally not an individual. Please revise your data statement to a non-author institutional point of contact, such as a data access or ethics committee, and send this to us via return email. Please also include contact information for the third party organization, and please include the full citation of where the data can be found.

5. Thank you for stating the following in your manuscript:

“The I-SOCIALISE trial was funded by the NIHR PHR programme. This paper was written after the completion of the grant for the trial and authors did not receive any additional funding for this paper.  “

Reviewers' comments:

Reviewer's Responses to Questions

**Comments to the Author**

1. Is the manuscript technically sound, and do the data support the conclusions?

Reviewer #1: Partly

2. Has the statistical analysis been performed appropriately and rigorously?

Reviewer #1: No

3. Have the authors made all data underlying the findings in their manuscript fully available?

Reviewer #1: Yes

4. Is the manuscript presented in an intelligible fashion and written in standard English?

Reviewer #1: Yes

Reviewer #1: [Title and Abstract]

This article details the implementation fidelity of LEGO® based therapy, which is implemented through school-based randomised controlled trials. This topic is important as it highlights the importance of process evaluation, which measures to what extent the intervention is implemented on the ground and offers insights for further improvement. In particular, stemming from the original impact evaluation study by (Wright et al., 2023), the current article focuses on the results from a small portion of the sample schools (9%) that participated in the independent fidelity assessment. However, the title and abstract have not addressed the scope and purpose of the current study and possibly misleading readers to expect a complete result of the impact evaluation. For example:

• The title and introduction need to clearly mention the implementation ‘fidelity’ as a core of the current study;

• RCT should not be highlighted, as the data used in this study only covers sub-groups of the treatment group;

• Methods need to be applied to the analytical sample of this study (22 schools participated in the independent fidelity assessment) rather than the entire sample from the original study. What method is used for the data analysis in the study, e.g., descriptive statistics?

[Introduction/ The Research Study]

• It may benefit from the detailed summary of Wright et al., (2023a) as a motivation for the current study.

• What is the age group of ‘young’ people from primary and secondary schools? Did the play-based approach adapt to age-appropriate practices to cover a wide range of age groups?

• (Trial Design and Delivery section) The programme was delivered within 12 weeks, but was there any reason to administer midline/endline assessments in 20 and 52 weeks after a few months from the programme implementation? The time gap between the intervention and assessment could explain the diminishing impact revealed in Wright et al. (2023a).

• (Table 1) The authors need to present this with the number of schools and students, as well as the distribution of treatment and control groups. However, it should be clear that this is not the sample of the following analysis in the results section.

• The purpose of the research and research questions need to be described explicitly. The first results on LEGO-based therapy delivery are descriptive statistics about the facilitator’s profile, previous knowledge or experience of autism/other interventions. The second results cover descriptive statistics of independent fidelity assessment through the rating consistency between facilitators and independent facilitators.

• (pg. 10 in PDF review file) The authors said, “Further details of the fidelity results are reported in Wright et al 2023b but does not include the analysis or discussion in this paper.” Please give more details about what you included and what you did not include in this paper.

• (pg. 13 in PDF review file)

- Why was self-reported fidelity analysed by sessions, not by facilitators?

- In total, 22 schools were included in the independent fidelity assessment – how were they selected? Had this group of schools any representativeness?

- You need to use headings and subheadings properly. For example, the facilitation of social interaction, structural quality, and adherence to the programmes should be under the analysis of trends.

- Also, it is not clear why this showed ‘trends’—was there any measurement of changes over time?

• (Table 6) “Fidelity rating by item and points of disagreements between facilitator and independent ratings are reported in Table 6.”

- The authors need to provide a description and interpretation of Table 6 in the results section.

- All items have 63 responses (but there are 62 responses in some rows without any missing variable – e.g., Did you highlight some aspect of social interaction and prompt children to problem solve or discuss it with each other?). How many facilitators and independent evaluators were included in this survey? It is unclear who are the independent trained researchers.

[Discussion]

• The overall article is very descriptive without probing the implementation fidelity more rigorously. For example, it would be more meaningful to see the relationship between the facilitator’s profile/experience and their score in the independent fidelity assessment (agreement/disagreement).

• The discussion section is rich in locating the findings in the literature. However, there are some redundant parts – e.g., three hours of training is too short. It could be more structured by implementation barriers that emerged from the study: knowledge or experience issues, time (or resource) constraints for training, etc.

• Although there is a separate limitations section, the authors listed limitations previously in the discussion section.

• Implications could be converged to conclusions.

**Do you want your identity to be public for this peer review?** For information about this choice, including consent withdrawal, please see our Privacy Policy

Reviewer #1: No

---

## [Author Response · Author response to Decision Letter 1]

15 May 2025

Response: Headings have been amended in line with guidance, double spacing has been used and there are no additional files.The second link did not work but I reviewed other guidance (tables) as required.

“I have read the journal's policy and the authors of this manuscript have the following competing interests: LEGO® is a registered trademark with a fair use policy which was adhered to throughout the duration of the trial. The LEGO® Foundation have agreed to the use of this term solely for this project and its outputs. Author GGC is a Director of Play Included, a community interest company that offers training and resources for interventions involving play bricks for children. Author MG is employed by Play Included, and JM worked for the company at the time of writing the paper.

Other authors report no conflict of interest.”

We note that one or more of the authors are employed by a commercial company: LEGO

Response: None of the authors currently work for the LEGO group. Three authors are affiliated to Play Included which has links to the LEGO foundation but the LEGO group does not have any influence over the company. We have updated the necessary statements to clarify this.

a. Funding statement has been updated as requested.

b. Competing interest statement has been updated to include:

JM, ML, GGC are employed by Play Included CIC, a not-for profit community interest company. Play Included have funding from the LEGO Foundation. The LEGO® Foundation is a Danish philanthropic organisation who fund charities and not-for-profit organisations to support learning through play. As such, Play Included has links to the LEGO® foundation but not the LEGO® Group (which is a commercial company that sells LEGO bricks). The LEGO group does not have any influence over Play Included. The LEGO Foundation was not involved in any part of the I-SOCIALISE trial or this paper. No commercial LEGO® entities, including the LEGO® group were involved in any part of the I-SOCIALISE trial, except granting special permission for the term LEGO® based therapy Play Brick Therapy to be used in research communications about the I-SOCIALISE trial (now known as Play Brick Therapy).

b) If there are no restrictions, please upload the minimal anonymized data set necessary to replicate your study findings to a stable, public repository and provide us with the relevant URLs, DOIs, or accession numbers. Please see http://www.bmj.com/content/340/bmj.c1ti1.long for guidelines on how to de-identify and prepare clinical data for publication. For a list of recommended repositories, please see https://journals.plos.org/plosone/s/recommended-repositories. You also have the option of uploading the data as Supporting Information files, but we would recommend depositing data directly to a data repository if possible.

Response: The trial sponsor (Leeds and York Partnership NHS Foundation Trust) has requested we do not share anonymised data sets due to the nature of consent provided by the participants and their guardians. They have agreed that we my be able to share data on request, and this will be assessed on a case-by-case basis.

The video recordings for the study have been destroyed following analysis and quality checking as per the ethics application. Again, we did not seek consent to share the videos outside of the trial team.

4. In the online submission form you indicate that your data is not available for proprietary reasons and have provided a contact point for accessing this data. Please note that your current contact point is a co-author on this manuscript. According to our Data Policy, the contact point must not be an author on the manuscript and must be an institutional contact, ideally not an individual. Please revise your data statement to a non-author institutional point of contact, such as a data access or ethics committee, and send this to us via return email. Please also include contact information for the third party organization, and please include the full citation of where the data can be found.

Response: We have amended this to the Quality Assurance team in Sheffield Clinical Trials Research Unit. The address is: ctruqa@sheffield.ac.uk

5. Thank you for stating the following in your manuscript:

“The I-SOCIALISE trial was funded by the NIHR PHR programme. This paper was written after the completion of the grant for the trial and authors did not receive any additional funding for this paper. “

Response: We have updated the Acknowledgement section to remove reference to funding.

“We acknowledge and thank the Chief Investigator of I-SOCIALISE, Professor Barry Wright and the full I-SOCIALISE team that successfully undertook and completed the trial:….”

6. Title and Abstract

This article details the implementation fidelity of LEGO® based therapy, which is implemented through school-based randomised controlled trials. This topic is important as it highlights the importance of process evaluation, which measures to what extent the intervention is implemented on the ground and offers insights for further improvement. In particular, stemming from the original impact evaluation study by (Wright et al., 2023), the current article focuses on the results from a small portion of the sample schools (9%) that participated in the independent fidelity assessment. However, the title and abstract have not addressed the scope and purpose of the current study and possibly misleading readers to expect a complete result of the impact evaluation.

Response: We have amended the title and added a sentence to the abstract background stating: ‘The main trial results are published elsewhere.’ We already mention fidelity in the methods section of the abstract.

7. The title and introduction need to clearly mention the implementation ‘fidelity’ as a core of the current study;

Response: Added to title. The introduction ends with a paragraph on fidelity, which we have added to (in bold), and now states:

“In this article, we report on the fidelity of LEGO® based therapy within a randomised controlled trial (see Varley et al. (2019) and Wright et al. (2023a) for the protocol and trial findings, and the NIHR report for more details, Wright et al. 2023b) and reflect on training and implementation of LEGO® based therapy in school settings. The trial found a small but non-clinically significant effect on the participants’ social skills and this paper explores potential reasons for the small effect based on the fidelity of the training and delivery of the intervention.”

We have added text after this around the research questions as per a comment below.

8. RCT should not be highlighted, as the data used in this study only covers sub-groups of the treatment group;

Response: We understand this comment and we do not want to be misleading, but we would like to keep the reference to the RCT as it is so important to the paper and is important in searching for trial methodology papers such as this one.

We hope the amended title make it clearer that we are using some data and our experience from the RCT.

9. Methods need to be applied to the analytical sample of this study (22 schools participated in the independent fidelity assessment) rather than the entire sample from the original study. What method is used for the data analysis in the study, e.g., descriptive statistics?

Response: We do still include data on all schools that delivered the intervention, it is only the independent assessment that was a sub sample. We have added the following sentence for clarification:

“Analysis

For this article, we are reporting on the data collected in the trial related to training facilitators in the intervention, self-reported fidelity for all intervention sessions (n=749) and a sub-sample (n=63) of sessions independently assessed for fidelity. All analyses are descriptive.”

10. It may benefit from the detailed summary of Wright et al., (2023a) as a motivation for the current study.

Response: We have added a paragraph reporting the main trial results and a sentence to the end of the introduction (as detailed above).

11. What is the age group of ‘young’ people from primary and secondary schools? Did the play-based approach adapt to age-appropriate practices to cover a wide range of age groups?

Response: 7-15 years. We have added this to the description of the trial.

Added to intervention description: The LEGO® sets (materials) were slightly different for 7-year-olds than 15-year-olds but otherwise training content and intervention delivery were similar/expected to be the same.

12. Trial Design and Delivery section: The programme was delivered within 12 weeks, but was there any reason to administer midline/endline assessments in 20 and 52 weeks after a few months from the programme implementation? The time gap between the intervention and assessment could explain the diminishing impact revealed in Wright et al. (2023a).

Response: We initially started with follow-up at 16 weeks to allow for delays in starting and breaks (for holidays or other reasons), but we amended this to 20 weeks early in the trial due to concerns that 16 weeks was not long enough and to avoid collecting data before the intervention had finished. Only one participant’s data was collected prior to the change.

The data collection at 52 weeks is to see if there was a longer term and lasting effect of the intervention.

We have added the following sentence to the trial description section:

“Participants were followed up 20 weeks and 52 weeks post baseline to identify the effects immediately after the intervention (whilst allowing for delays in starting and breaks) and longer-term effects of the intervention.”

13. Table 1: The authors need to present this with the number of schools and students, as well as the distribution of treatment and control groups. However, it should be clear that this is not the sample of the following analysis in the results section.

Response: We have added a statement around the data included in the paper at the end of the following sentence (in bold):

In total, 98 schools participated in the I-SOCIALISE trial (see Table 1 for further details), 50 schools were allocated to deliver LEGO® based therapy, with 127 young people and 82 trained facilitators in total; this paper only reports on the intervention delivery and does not include information from the 48 control schools.

We have also updated Table 1 to include the requested data.

14. The purpose of the research and research questions need to be described explicitly. The first results on LEGO-based therapy delivery are descriptive statistics about the facilitator’s profile, previous knowledge or experience of autism/other interventions. The second results cover descriptive statistics of independent fidelity assessment through the rating consistency between facilitators and independent facilitators.

Response: Two research questions have been added to the end of the introduction section: 1. How did the implementation of the training and delivery of LEGO® based therapy in school settings impact on the results of the trial? 2. What lessons can we learn to improve the implementation of the programme?

15. (pg. 10 in PDF review file) The authors said, “Further details of the fidelity results are reported in Wright et al 2023b but does not include the analysis or discussion in this paper.” Please give more details about what you included and what you did not include in this paper.

Response: Some text added: “Wright et al 2023b includes details of the number of sessions delivered and the components delivered based on the facilitator checklist and the independent assessors, and this paper includes additional detail on specific findings from the fidelity assessment, information about the facilitators and discussion aro

---

## [Decision Letter · Decision Letter 1]

11 Jul 2025

Dear Dr. Biggs,

Thank you for submitting your manuscript to PLOS ONE. After careful consideration, we feel that it has merit but does not fully meet PLOS ONE’s publication criteria as it currently stands. Therefore, we invite you to submit a revised version of the manuscript that addresses the points raised during the review process.

We look forward to receiving your revised manuscript.

Kind regards,

Yuliang Zhang, Ph.D.

Academic Editor

PLOS ONE

Reviewers' comments:

Reviewer's Responses to Questions

**Comments to the Author**

Reviewer #2: All comments have been addressed

Reviewer #3: (No Response)

2. Is the manuscript technically sound, and do the data support the conclusions?

Reviewer #2: Yes

Reviewer #3: Partly

3. Has the statistical analysis been performed appropriately and rigorously?

Reviewer #2: Yes

Reviewer #3: No

4. Have the authors made all data underlying the findings in their manuscript fully available?

Reviewer #2: No

Reviewer #3: No

5. Is the manuscript presented in an intelligible fashion and written in standard English?

Reviewer #2: Yes

Reviewer #3: Yes

Reviewer #2: Based on the detailed revisions and responses in the current version of the manuscript, the authors have adequately addressed the comments raised in the previous review round. They have clarified the scope and purpose of the study, emphasizing fidelity rather than impact evaluation results.The title, abstract, introduction, and methods have been revised to properly reflect the subsample analysis, descriptive nature, and focus on implementation fidelity.Additional methodological detail and clarifications were provided regarding sample sizes, data handling, and assessment procedures.The Discussion section has been significantly improved with better structure and clearer delineation of implementation barriers and recommendations.Data availability, funding, and competing interest statements have been corrected and aligned with journal policies. The manuscript presents a rigorous, well-reported fidelity analysis of a complex intervention in real-world school settings. The methods and data appropriately support the conclusions. The study contributes valuable insights into the implementation of play-based social skills interventions and meets PLOS ONE’s standards for technical soundness.

While the manuscript is solid and suitable for publication, there are several specific suggestions that could further strengthen the paper from scientific, practical, and readability perspectives:Clarify Sampling Representativeness, Add a sentence on whether these schools differed in any systematic way from the others (e.g., school type, facilitator experience), or clarify that such comparison was not conducted;Propose a preliminary conceptual framework or logic model that includes facilitator behaviors, child engagement, and social outcomes — even if exploratory.

Reviewer #3: There is growing evidence that cultural factors can influence the presentation and interpretation of autistic traits (e.g., see https://pubmed.ncbi.nlm.nih.gov/39882425/). Given that LEGO® is a globally recognized product and this paper is likely to be read by an international audience, the authors are strongly encouraged to provide a clearer explanation of how the study’s conclusions based on data from a specific geographic region, can be interpreted in broader, cross-cultural contexts. Specifically, it would be helpful to clarify whether any diversity-related variables such as race, ethnicity, or country of origin were considered in the analysis or participant characterization.

Many of the conclusions in the manuscript are drawn from a subset of the data, with only 9% of sessions independently reviewed for fidelity. This limitation affects the generalizability of the findings and should be emphasized more clearly in both the discussion and limitations sections.

Furthermore, discrepancies observed between facilitator self-reports and independent ratings, particularly in areas such as role discussion and reward usage, raise concerns about potential overreporting of fidelity. These inconsistencies merit further reflection, particularly in the context of assessing the reliability of self-reported implementation data. It would also be beneficial to explain why fidelity was analyzed on a per-session basis rather than per facilitator, and to discuss what limitations this approach may impose on interpreting trends over time or individual consistency in program delivery.

In Table 6, the authors use “N/A” to denote certain responses. Further clarification on how these responses were treated in percentage calculations would be appreciated to ensure interpretive clarity. While descriptive statistics are appropriate for a study of this nature, the lack of even basic cross-tabulations or correlation analyses slightly weakens the analytical depth. Incorporating simple exploratory statistics, for example, examining the relationship between facilitator experience and fidelity ratings would strengthen the empirical rigor.

Since the main trial reported only small effects, the authors should more clearly articulate how the fidelity challenges identified in this paper, particularly those related to social interaction facilitation, may have contributed to the limited intervention impact.

**Do you want your identity to be public for this peer review?** For information about this choice, including consent withdrawal, please see our Privacy Policy

Reviewer #2: No

Reviewer #3: **Yes:**  Sankar Raju Narayanasamy

---

## [Author Response · Author response to Decision Letter 2]

23 Oct 2025

1, #1 Clarify Sampling Representativeness, Add a sentence on whether these schools differed in any systematic way from the others (e.g., school type, facilitator experience), or clarify that such comparison was not conducted

Response: We have added that this comparison was not conducted.

“To be eligible for this part of the study, all group participants and facilitators needed to consent to be video recorded, so where that was in place, the study team approached the relevant schools to organise video recording; we did not assess the representativeness of the schools that participated in the independent fidelity assessment.”

1, #2 Propose a preliminary conceptual framework or logic model that includes facilitator behaviors, child engagement, and social outcomes — even if exploratory.

Response: We have developed a preliminary model and added as an appendix.

We have added the following text to the discussion:

“It's important to note that trials of complex interventions increasingly emphasise using a clear theory of change and logic model to assess intervention effectiveness, helping to identify factors influencing when and why the intervention works, a pre-liminary logic model is included in the appendix based on the delivery in the trial and the trial findings”

2, #1 There is growing evidence that cultural factors can influence the presentation and interpretation of autistic traits (e.g., see https://pubmed.ncbi.nlm.nih.gov/39882425/).

Given that LEGO® is a globally recognized product and this paper is likely to be read by an international audience, the authors are strongly encouraged to provide a clearer explanation of how the study’s conclusions based on data from a specific geographic region, can be interpreted in broader, cross-cultural contexts. Specifically, it would be helpful to clarify whether any diversity-related variables such as race, ethnicity, or country of origin were considered in the analysis or participant characterization.

Response: Added the following text to the limitations section:

“LEGO® is a globally recognised toy and there is evidence of cross-cultural differences in Autism (Martinez-Gonzalez et al., 2025) but we are unable to comment on this as we did not collect data on county of origin, and the sample was not diverse enough for us to explore cultural differences. However, there has been some evaluation of the intervention in Australia, Denmark, Ireland, Kenya, Malta, Mexico, and the US where is was feasible to deliver and acceptable to children. The qualitative study found positive changes in children’s enjoyment of being in a group, confidence, friendship and language development, but there was no impact on anxiety, emotional regulation or understanding of the child’s own feelings (Laurie and Gomez de la Cuesta, 2025).”

2, #2 Many of the conclusions in the manuscript are drawn from a subset of the data, with only 9% of sessions independently reviewed for fidelity. This limitation affects the generalizability of the findings and should be emphasized more clearly in both the discussion and limitations sections.

Response: Added a sentences to make this clear in the discussion:

“We measured fidelity in two ways: through self-report and through independent assessment of 9% of sessions.”

And a sentence to the to the limitations section:

“….and only 9% of the sessions were independently assessed for fidelity, so most of our findings are based on self-reported fidelity.”

2, #3 Furthermore, discrepancies observed between facilitator self-reports and independent ratings, particularly in areas such as role discussion and reward usage, raise concerns about potential overreporting of fidelity. These inconsistencies merit further reflection, particularly in the context of assessing the reliability of self-reported implementation data.

Response: We have added some text to the limitations around this:

“The main measure of fidelity was by self-report checklists completed by the facilitators, where there is potential for overreporting fidelity (Borelli, 2011) and can be unreliable due positivity bias (Hansen et al., 2014). This was evident when compared to the independent assessments, but agreement between our self-report and independent assessments were generally high and we did not have the resources to independently assess every session.”

2, #4 It would also be beneficial to explain why fidelity was analyzed on a per-session basis rather than per facilitator, and to discuss what limitations this approach may impose on interpreting trends over time or individual consistency in program delivery.

Response: We had already stated this in the methods:

“Fidelity was analysed by session to understand fidelity over the course of the 12 weeks, and it was deemed more logical to approach the analysis in this way.”

We have also added the following to the limitations:

“The independent assessment was conducted on a per-session basis, not on a per-facilitator basis. Although 22 facilitators were assessed over two or more sessions throughout the 12-week intervention, this approach makes it challenging to attribute the fidelity findings to specific facilitators' experience or characteristics. This, in turn, makes it difficult to pinpoint individual training needs or areas for improvement for practitioners in future interventions.”

2, #5 In Table 6, the authors use “N/A” to denote certain responses. Further clarification on how these responses were treated in percentage calculations would be appreciated to ensure interpretive clarity. While descriptive statistics are appropriate for a study of this nature, the lack of even basic cross-tabulations or correlation analyses slightly weakens the analytical depth.

Response: We have added this as a footnote to the table:

“The denominator for working out the percentages is 63 (the number of sessions that were independently assessed).”

2, #6 Incorporating simple exploratory statistics, for example, examining the relationship between facilitator experience and fidelity ratings would strengthen the empirical rigor.

Response: We have added two tables breaking down the fidelity ratings by facilitator experience and added some text around this.

Table 6 text: “For 71 facilitators (those who delivered the intervention in the trial and completed self-report forms), we have looked at the level of autism knowledge and their experience delivering interventions for autism against their self-report ratings. Table 6 below shows the average fidelity ratings across the 4 core items, and for all items by their level of experience. This is an average rating and does not account for trends over time. The self-report rating on all items does not appear linked to the facilitators experience, with moderate and sound experience leading to the highest self-report ratings and those delivering 0 interventions scoring higher on the self-report ratings as well. However, for the 4 core items on the checklist, there appears to be an increase in self-reported fidelity with an increased in knowledge of autism, but there was no clear pattern with increased numbers of interventions delivered previously.”

Table 7 text: “Again, we looked at the independent fidelity ratings according to the facilitators experience (N= 15, lower than the number delivering due to missing data or because there was more than one facilitator present). Table 7 shows the independent fidelity ratings for all recorded sessions we could match with a unique facilitator by the level of experience of the facilitator, and this show that on both experience measures, the ratings were slightly higher where facilitators were more experienced.”

Text added to discussion: “There was some evidence of this when looking at the independent fidelity ratings by facilitator experience, as fidelity ratings were slightly higher for facilitators with more experience (Table 7). Although there was an indication of this in the self-report for the 4 core fidelity items, this was not shown in the self-report ratings across all fidelity items. It should be noted that these are small numbers, and the nature of self-reported fidelity, which could also vary by experience, means we should not draw firm conclusions based on these numbers.”

2, #7 Since the main trial reported only small effects, the authors should more clearly articulate how the fidelity challenges identified in this paper, particularly those related to social interaction facilitation, may have contributed to the limited intervention impact.

Response: Summary paragraph added in the discussion.

“In summary, the I-SOCIALISE trial observed a small, non-clinically significant improvement in participants' social skills, which we believe is due to challenges in the fidelity of the intervention's delivery, particularly regarding social interaction facilitation. While structural elements like collaborative building were largely adhered to, facilitators, who had variable prior experience with autism and seemed to underestimate their crucial role in guided play, tended to focus more on completing LEGO® builds rather than actively fostering and reinforcing social interaction and positive relationships among the young people. This was exacerbated by the limited three-hour training duration, which likely prevented a deeper understanding of the facilitator role in supporting social development through play, and challenges in recognising and highlighting positive social interactions for autistic participants. Consequently, the fidelity gaps in facilitating the core social aims of the therapy are considered key contributors to the intervention's limited impact on social skills, underscoring the need for future training to prioritise active guided play and understanding neurodivergent communication styles.”

---

## [Editor Report · Decision Letter 2]

2 Nov 2025

Lessons learnt about implementing LEGO® based therapy (Play Brick Therapy) based on fidelity data and experience from a large school-based randomised controlled trial.

PONE-D-24-35083R2

Dear Dr. Biggs,

We’re pleased to inform you that your manuscript has been judged scientifically suitable for publication and will be formally accepted for publication once it meets all outstanding technical requirements.

Kind regards,

Yuliang Zhang, Ph.D.

Academic Editor

PLOS ONE
---

## [Editor Report · Acceptance letter]

PONE-D-24-35083R2

PLOS One

Dear Dr. Biggs,

I'm pleased to inform you that your manuscript has been deemed suitable for publication in PLOS One. Congratulations! Your manuscript is now being handed over to our production team.

Kind regards,

on behalf of

Dr. Yuliang Zhang

Academic Editor

PLOS One